# Isolation and Identification of Herbicidal Active Compounds from *Brassica oleracea* L. and Exploration of the Binding Sites of Brassicanate A Sulfoxide

**DOI:** 10.3390/plants12132576

**Published:** 2023-07-07

**Authors:** Yu Wang, Wanyou Liu, Baozhu Dong, Dong Wang, Yin Nian, Hongyou Zhou

**Affiliations:** 1College of Horticulture and Plant Protection, Inner Mongolia Agricultural University, Hohhot 010020, China; wang_style@emails.imau.edu.cn (Y.W.); nongda@emails.imau.edu.cn (W.L.); wangdong2002@imau.edu.cn (D.W.); 2State Key Laboratory of Phytochemistry and Plant Resources in West China, Kunming Institute of Botany, Chinese Academy of Sciences, Kunming 650201, China

**Keywords:** *Brassica oleracea* L., activity guided, allelochemicals, herbicidal activity, molecule docking

## Abstract

*Brassica oleracea* L. has strong allelopathic effects on weeds. However, the allelochemicals with herbicidal activity in *B. oleracea* L. are still unknown. In this study, we evaluated the activity of allelochemicals isolated from *Brassica oleracea* L. based on the germination and growth of model plant *Lactuca sativa* Linn., grass weed *Panicum miliaceum*, and broadleaf weed *Chenopodium album.* Additionally, we employed molecular docking to predict the binding of brassicanate A sulfoxide to herbicide targets. The results of this study showed that eight compounds with herbicidal activity were isolated from *B. oleracea* L., and the predicted results indicated that brassicanate A sulfoxide was stably bound to dihydroxyacid dehydratase, hydroxymethylpyruvate dioxygenase, acetolactate synthase, PYL family proteins and transport inhibitor response 1. This research provides compound sources and a theoretical foundation for the development of natural herbicides.

## 1. Introduction

Weeds, one of the three primary agricultural pests, result in substantial yield losses. Therefore, significant endeavors have been undertaken to devise innovative approaches aimed at weed control and management [1,2]. A common weed control measure is the spraying of chemical herbicides, which are used to prevent weed damage and thus protect farmers’ incomes. Throughout the history of agriculture, synthetic compounds have been powerful tools for controlling weeds and pests. However, the use of synthetic herbicides has caused a range of ecological problems, including excessive herbicide residues in soil and food, which are harmful to farmland ecosystems and mammals [3,4,5]. Moreover, the improper use of herbicides has contributed to the emergence of weed resistance, thereby exacerbating the associated damages [6,7]. Therefore, there is an urgent need to develop safe and effective herbicides for the sustainable development of agriculture.

At present, herbicides commonly used are 4-hydroxyphenylpyruvate dioxygenase (HPPD) inhibitors, dihydroxyacid dehydratase (DHAD) inhibitors and acetolactate synthase (ALS) inhibitors [8,9]. In addition, there are plant growth regulators that target the strigolactone receptor hydrolase D14-D3-ASK1, PYL family proteins and transport inhibitor response 1 [10,11,12].

The ability of plants to influence the growth and development of other plants and organisms through the production and release of secondary metabolites is known as allelopathy, and these secondary metabolites are referred to as allelochemicals [13,14]. As an allelopathic plant, *B. oleracea* L. contains a variety of allelopathic chemicals that can inhibit a wide range of microorganisms and plants when released into the soil [15,16]. When using *B. oleracea* L. residue as field cover, its allelopathic effect can significantly reduce the frequency of weed occurrence and weed biomass in the field [17]. Allelopathy can be used as a complement to chemical weed control measures to reduce the use of chemical herbicides. There is great potential to develop the allelopathic chemicals of *B. oleracea* L. as a green pesticide to use for weed control [18,19,20]. However, the allelochemicals of *B. oleracea* L. with herbicidal activity and their targets remain unclear.

The aim of our research is to isolate and identify herbicidal active compounds from *B. oleracea* L. Moreover, we hypothesize that active compounds will interact with known herbicide targets such as HPPD, ALS, DHAD, PLY2, D14-D3-ASK1 and TIR1. These identified active compounds hold the potential to serve as a basis for the development of novel and environmentally friendly herbicides. Additionally, they could play a crucial role in mitigating the damage caused by herbicide-resistant weeds.

## 2. Results

### 2.1. Extraction and Separation Processes

A powder of *B. oleracea* L. was subjected to extraction using 75% ethanol. Subsequently, components with similar polarity were combined based on the results of thin-layer chromatography (Appendix A). Five fractions (1–5) were separated by trichloromethane/MeOH column chromatography and the herbicidal activity of the fractions was assessed by germination and development tests in Petri dishes on *L. sativa* Linn. All fractions obtained from the crude extracts inhibited the growth of roots and stems of *L. sativa* Linn. seedlings. *L. sativa* Linn. is an ideal dicotyledonous plant for preliminary studies as it can be used for the microbiological testing of studies when few compounds are tested, as is the case with natural compounds [21]. Comparing the inhibition rates of the five fractions on *L. sativa* Linn. seedlings, Fr 2 showed the highest inhibition of root elongation in *L. sativa* Linn. seedlings, with a root length of 0.86 cm. However, all fractions inhibited the stems of *L. sativa* Linn., with no significant differences between all fractions (Figure 1A–C).

The Fr 2 fraction was separated by normal phase column chromatography (silica gel) using a mixture of PE:EtOAc to increase polarity. The subfractions with similar polarity were merged based on the results of thin-layer chromatography (Appendix A). Six subfractions of interest (1–6) were obtained and assessed by Petri dish germination and development tests on *L. sativa* Linn. The results showed that three subfractions (Fr 2.3, Fr 2.4 and Fr 2.5) significantly inhibited the roots and stems of *L. sativa* Linn., with Fr 2.3 and Fr 2.4 having stronger inhibitory effects on the roots, which shortened to 0.76 cm and 0.66 cm under the influence of the two subfractions (Figure 1D–F). Therefore, the Fr 3 and Fr 4 subfractions continued to be isolated for further experiments.

### 2.2. Isolation of Allelochemicals

Eight compounds (1–8) were obtained by further isolation and tested against *L. sativa* Linn. in Petri dishes. Compound 1 inhibited the germination of *L. sativa* Linn. seeds by only 13.3%, while compounds 2 and 4 also caused a significant reduction in the germination of *L. sativa* Linn. seeds (16.6% and 20.0%). Compounds 1, 2 and 4 were not significantly different than the positive agent Tribenuron Methyl^®^ and all reached the level of herbicide action (Figure 2G,H). The effects of these compounds in *B. oleracea* L. were close to those achieved by commercial herbicides. For weeds, we propose a possibility: a new strategy for plant protection.

The root and stem lengths of the *L. sativa* Linn. seedlings barely elongated under the effects of compound 1. The root elongation of the *L. sativa* Linn. seedlings was strongly inhibited by compounds 1, 2, 3 and 4. The inhibition of *L. sativa* Linn. root was 38.59% under the effect of compound 4 and 23.04% under the effect of compounds 2 and 3. The inhibition rates of *L. sativa* Linn. seedling stem elongation by compounds 1, 2, 3 and 4 were close to the effect of positive control (Figure 2A–C). The eight compounds isolated in this study inhibited both root and stem elongation of *L. sativa* Linn., but to different extents.

Eight compounds had a significant inhibitory effect on the elongation of *C. album* stems, with all except compound 5 approaching the effect of the positive control. However, the root elongation was only affected by compounds 1, 2, 3 and 4, with only compounds 1 and 4 approaching the effect of the positive herbicide (Figure 3A–E). The shoots of *P. miliaceum* were strongly inhibited by compound 1 and did not differ from the positive control. However, the root elongation of *P. miliaceum* was still significantly affected by compounds 1, 2, 3 and 4, with compounds 1 and 4 not differing from the levels of the positive herbicide (Figure 3F–I).

The identities of the indole skeleton compounds (1–5), aldehyde (6), diterpenoid (7) and organic acid (8), were determined by comparing their spectral properties (MS, ^1^H NMR, ^13^C NMR and specific rotation). In this way, compound 1, the main product of the active Fr 2.3, was isolated as a yellow solid (methanol) and identified as indole-3-carboxaldehyde (C_9_H_7_NO). There was a minor amount of compound 2 in Fr 2.3, in the form of a yellow oil-like substance, which was identified as (-)-Spirobrassinin (C_11_H_10_N_2_OS_2_). A very small amount of compound 3, as yellow needle-like crystals (methanol), was identified as brassicanate A sulfoxide (C_11_H_11_NO_3_S). Compound 4 was the main product of Fr 2.4, as yellow crystals (methanol) identified as methyl 1H-indole-3-carboxylate (C_10_H_9_NO_2_). Compound 5 was ethyl 6-amino-2-methyl-1H-indole-3-carboxylate (C_12_H_14_N_2_O_2_), a yellow oil-like compound (methanol). Compounds 6, 7 and 8 were isolated from Fr 2.2 and were identified as 4-Hydroxybenzaldehyde (C_7_H_6_O_2_), diisobutyl phthalate (C_16_H_22_O_4_) and (10 E, 12 E)-9-Oxo-10, 12-octadecadienoic acid (C_18_H_30_O_3_) (Figure 4).

### 2.3. Docking of Brassicanate A Sulfoxide into the Active Site of Herbicide

The binding modes of the separated brassicanate A sulfoxide to the herbicide targets DHAD, HPPD and ALS crystals were established by autodock simulations. The binding energies of brassicanate A sulfoxide to the target proteins were less than −5 Kcal/mol. Gaillard has concluded that the docking effect was good when the binding energy was below −5 Kcal/mol [22]. Therefore, brassicanate A sulfoxide had a good binding affinity for DHAD, HPPD and ALS. The lowest binding energy of brassicanate A sulfoxide to DHAD was −6.19 Kcal/mol, indicating a higher possibility of an interaction between brassicanate A sulfoxide and DHAD. The brassicanate A sulfoxide was bound to DHAD through one hydrogen bond, indicating that brassicanate A sulfoxide and DHAD can form a stable conformation with high binding stability (Figure 5).

### 2.4. Docking of Brassicanate A Sulfoxide into the Active Site of Plant Growth Regulator

Among the targets of a plant growth regulator, TIR1 acts as a receptor protein for auxin, regulating the growth and development of plant roots [12]. The molecular docking energy of brassicanate A sulfoxide with TIR1 was −6.22 Kcal/mol, indicating that brassicanate A sulfoxide formed a stable conformation with TIR1 through hydrogen bonding. The natural substrate of PLY2-HAB1 is abscisic acid, and it exhibits binding energies greater than −5 Kcal/mol when molecular docking is performed using brassicanate A sulfoxide as a ligand. The binding energy of brassicanate A sulfoxide with D14-D3-ASK1 is 61.67 Kcal/mol. However, brassicanate A sulfoxide cannot form a stable structure with the plant growth-regulating target protein D14-D3-ASK1 (Figure 6).

### 2.5. Effects of (-)-Spirobrassinin and Brassicanate A sulfoxide on ALS, DHAD and HPPD

In order to verify the predictions of docking, we assayed the activity levels of the target proteins in *L. sativa* Linn. seedlings. The results showed that brassicanate A sulfoxide reduces the activity of ALS, DHAD and HPPD in *L. sativa* Linn. seedlings (Figure 7).

## 3. Discussion

In response to weed damage, commercial synthetic herbicides are often used to intervene in agricultural production [23]. At the same time, the overuse of commercial herbicides has revealed many problems, such as excessive pesticide residues and pollution of farmland ecosystems [24]. The application of natural products with herbicidal activity in production has become a new idea for controlling weeds in agricultural fields—for example, the use of benzoxazinoids to control weeds in wheat fields [25]. To address the problem of weed control in agricultural production, the compounds with herbicidal activity in *B. oleracea* L. were isolated and molecular docking was performed to find possible targets for allelochemicals in *B. oleracea* L.

In this study, indole-3-carboxaldehyde and (-)-Spirobrassinin had strong inhibitory effects on plant growth. Indole-3-carboxaldehyde is a plant growth factor whose regulatory effects on plant growth have been well documented, and indole-3-aldehyde plays an important role in the apical dominance of pea seedlings as a lateral bud-growth inhibitor [26]. (-)-Spirobrassinin and brassicanate A sulfoxide are allelochemicals unique to the cruciferous family and are characterized by the presence of sulfur atoms and indole groups [27]. Members of the Brassicaceae family contain glucosinolates—sulfur-containing molecules that are hydrolyzed to form compound allelopathy in a variety of soil-borne organisms, including weeds [15]. Glucosinolates consist of a glucose molecule, a sulfur moiety and a side chain whose composition determines their properties [28,29]. Glucosinolate molecules are not activated but are enzymatically hydrolyzed to produce a variety of biologically active products, including isothiocyanates, ionic thiocyanates, nitriles, oxazolidinethiones, organic cyanates and epithionitriles [30].

The results of the Petri dish germination and growth test showed that compounds 1, 2, 3, 4 and 8 inhibited the growth of *L. sativa* Linn. and weeds more strongly than the other compounds at a concentration of 0.25 mg/mL. Interestingly, a common feature of the compounds with good inhibitory activity is the presence of an indole group, which is the most characteristic feature of the growth hormone analogues [31]. There has not been any definitive report on the ability of indolic compounds, other than indole-3-acetic acid (IAA), to regulate plant growth. IAA is the natural plant growth hormone known to have the ability to regulate plant growth and development [32]. Further research is needed to determine whether the other indolic compounds isolated in our study are intermediate products in the synthesis of indoleacetic acid. The difference in the intensity of the inhibitory effect of these compounds on plant growth may be due to the differences in the substituent groups on the indole group [33,34]. The allelochemical compound 4-hydroxybenzaldehyde has a significant inhibitory effect on cucumber seedlings [35], while the remaining compounds had a weaker effect on *L. sativa* Linn. and weeds. Despite this, the results were statistically different, as can be easily seen by the observation.

In the Brassicaceae family, there are self-generated compounds containing sulfur atoms and indoles. Interestingly, we have discovered that (-)-Spirobrassinin possesses novel herbicidal activity, in addition to its known anti-cancer cell proliferation and anti-leukemic effects [36,37,38]. Molecular docking studies of brassicanate A sulfoxide, with herbicide targets and plant growth-regulating targets, have revealed an interesting finding that brassicanate A sulfoxide can stably bind to multiple targets such as DHAD, HPPD, ALS, PLY2-HAB1 and TIR1. We predict that brassicanate A sulfoxide may be a multi-target compound.

Brassicanate A sulfoxide cannot form a stable conformation with D14-D3-ASK1, but it can form stable conformations when interacting with other herbicidal targets and plant growth regulatory targets. HPPD catalyzes the conversion of 4-hydroxyphenylpyruvic acid (HPPA) by adding two oxygen atoms to the HPPA molecule, resulting in the formation of homogentisic acid (HGA) [39]. The inhibition of HPPD results in a decrease in plastid quinone and α-tocopherol, which leads to the protection of chlorophyll molecules and cystoid membranes from oxidative degradation [40]. The biosynthetic pathway of branched-chain amino acids (BCAA) is essential for plant growth, and the BCAA biosynthetic pathway in plants is carried out by three enzymes: acetolactate synthase (ALS), acetohydroxy acid isomeroreductase (KARI) and dihydroxyacid dehydratase (DHAD) [41,42]. It is worth noting that there are few herbicides that target both DHAD and KARI enzymes. The effects of brassicanate A sulfoxide on *L. sativa* Linn. showed that *L. sativa* Linn. seedlings had a normal green color with no symptoms of bleaching (Appendix A). Therefore, HPPD as an herbicide locus was excluded with high probability. The growth phenotype of *L. sativa* Linn. suggests that brassicanate A sulfoxide may act as an inhibitor of the biosynthesis of essential amino acids in plants, thus causing growth defects in *L*. *sativa* Linn. (Figure 2). Therefore, further studies are required to determine the specific targets of brassicanate A sulfoxide to verify that brassicanate A sulfoxide has an inhibitory effect on the heterologous expression of these target proteins in *Escherichia coli*. In addition, the HPPD inhibitor can cause early bleaching of plant leaves [43,44], but our experimental results did not reveal such phenomena. Therefore, the inhibitory effect of brassicanate A sulfoxide on *L. sativa* Linn. and weeds is likely to be inhibition of the biosynthesis of essential amino acids, which affects the normal growth and development of the plant.

In this study, a total of five indole skeleton compounds, one aldehyde compound, one organic acid compound, and one diterpenoid compound were isolated from *B. oleracea* L. Their planar and stereochemical structures were determined through 1D-NMR and 2D-NMR analysis. Some of these compounds exhibited significant inhibitory effects on three plant species: *L. sativa* Linn., *P. miliaceum* and *C. album*. Furthermore, brassicanate A sulfoxide exerted herbicidal activity by affecting the activity of DHAD, HPPD and ALS enzymes. This study not only enhances the application prospects of allelopathic effects of *B. oleracea* L. but also supports a promising strategy of developing natural product herbicides.

## 4. Materials and Methods

### 4.1. General

Both 1D and 2D NMR spectra were recorded on a Bruker Ascend 600 MHz spectrometer (Bruker Corp., Fällanden, Switzerland) with TMS internal standards. The chemical shifts were given in ppm for the ^1^H residual signals; the MeOH-d1 (δ 3.31), acetone-d1 (δ 2.05) and the ^13^C signals refer to solvent signals (δ 29.84) and (δ 49.00), respectively. The thin-layer chromatography (TLC) was performed on TLC Silica gel 60 F254 aluminum foil and TLC Silica gel 60 RP-18 F254S aluminum foil from Merck (Darmstadt, Germany). Semi-preparative HPLC was performed on a Waters e2695 liquid chromatography (Waters, Camden, NJ, USA) with a Zorbax SB-C18 (9.4 × 250 mm) column. HRESIMS data were obtained on an Agilent 6540 QSTAR TOF time-of-flight mass spectrometer (Agilent Corp., Palo Alto, CA, USA). Column chromatography (CC) was performed with silica gel (200–300 mesh; Qingdao Marine Chemical, Inc., Qingdao, China), MCI gel (75–150 μm, Mitsubishi Chemical Corporation, Tokyo, Japan). Fractions were monitored by thin-layer chromatography with spot detection in EtOH using 10% H_2_SO_4_.

### 4.2. Plant Material

The stems and leaves of *B. oleracea* L. were collected in September 2020 in Hohhot, Inner Mongolia Autonomous Region, China (44°08′ N, 111°41′ E), dried at 60 °C and then powdered for later analysis.

### 4.3. Organic Solvents

The methanol (MeOH), trichloromethane (TCM), petroleum ether (PE), ethyl acetate (EtOAc) and acetonitrile (AcN) (Hipersolv Chromanorm for HPLC) were purchased from VWR International (Radnor, PA, USA). MagniSolv acetone-d1 (99.8% deuteration degree) and methanol-d1 (99.8% deuteration degree) were used for NMR spectroscopy.

### 4.4. Extraction and Isolation

The samples obtained in 4.2 (2 kg) were extracted with 75% EtOH at room temperature to give crude extracts, which were successively decolorized on MCI gel with MeOH/H_2_O (90:10) to obtain extracts (800 g). The separation was then carried out by column chromatography using a TCM/MeOH mixture with increasing polarity from 0 to 100%. The following five subfractions were obtained: Fr 1 (4.7 g), Fr 2 (38 g), Fr 3 (67 g), Fr 4 (108 g) and Fr 5 (286 g). These subfractions were subjected to *Lactuca sativa* bioassays and all were confirmed to be active. The active fraction of Fr 2 was separated by chromatography.

Fr 2 was separated by column chromatography using a PE/EtOAc solvent mixture at increasing polarity, obtaining several subfractions: Fr 2.1 to Fr 2.6. Fr 2.2, Fr 2.3 and Fr 2.4 had the best activity and were again separated by column chromatography using a PE/EtOAC solvent mixture (2:1). The fractions obtained were again subjected to HPLC (reverse-phase semi-preparative column) using an AcN/H_2_O solvent mixture (5%: 95% *v*/*v*, flow rate: 3 mL/min), with the AcN ratio increasing linearly to 95% within 30 min. After purification, Fr 2.4 gave rise to compound 1 (62 mg), compound 2 (53.4 mg) and compound 3 (16.5 mg). Fr 2.3 gave rise to compound 8 (62 mg), compound 6 (53.4 mg) and compound 7 (16.5 mg). Fr 2.2 (0.7363 g) was passed through by HPLC (reverse-phase semi-preparative column) using an AcN/H_2_O solvent mixture for purification in order to yield compound 4 and compound 5.

### 4.5. Seed Germination and Growth Tests

The plant species selected for the experimental study were the model plant *L. sativa* Linn., the grass weed *P. miliaceum*, and the broadleaf weed *C. album*. *L. sativa* Linn. germination and seedling growth tests were carried out on fractions, subfractions and products. This test was commonly used to assess the sensitivity of *L. sativa* Linn. to biologically active substances. Full and uniformly mature *L. sativa* Linn. seeds, *C. album* and *P. miliaceum* were sterilized using the filter paper method in Petri dishes, soaked in a 0.2% sodium hypochlorite solution for 10 min and then washed three times with distilled water. Two layers of sterile filter paper were placed at the bottom of the Petri dishes and three replicates were set up by taking 20 seeds and scattering them evenly in the Petri dishes (diameter = 60 mm) with an equal amount of distilled water as a control. These seedlings were grown at a constant temperature of 25 ± 0.5 °C with 85% humidity and a 12 h light/12 h dark cycle. The number of germinated seeds was counted from 2 d after treatment for one week. After 7 d, the root length and shoot length of the seedlings were measured. The germination rate was calculated as: germination rate = number of germinated seeds/total number of seeds × 100%.

Samples for the tests were dissolved in distilled water (samples insoluble in water were heated to help them dissolve). Extracts, fractions and compounds were used at a concentration of 0.25 mg/mL and each test was carried out using 2 mL of extract, compound or Tribenuron Methyl^®^, with three replicates per sample.

### 4.6. Molecular Docking

The pdb format files for herbicide targets were retrieved from the protein structure database via PDB (https://www.rcsb.org/, accessed on 14 January 2022), ALS (PDB code: 1OZF), DHDA (PDB code: 5ZE4) and HPPD (PDB code: 6J63), plant growth regulatory targets: PLY2-HAB1 (PDB code: 3KDI), PLY10-PP2C (PDB code: 3RT0), D14-D3-ASK1 (PDB code: 5HZG) and TIR1 (PDB code: 2P1Q) within the download allelochemical: brassicanate A sulfoxide structure in the sdf format from the PubChem database (https://pubchem.ncbi, accessed on 14 January 2022). The AutoDock tool was used to add data on essential hydrogen atoms, and the Autogrid program was used to generate affinity (grid) maps of X:58 Y:60 Z:62 Å grid points with a spacing of 0.375 Å. The lamarckian genetic algorithm (LGA) and the Solis and Wets local search method were used to generate the docking.

The initial orientation, position and torsion of the ligand molecule were set randomly. Ten different runs were used to derive the results of the docking experiment. The docking results were saved as pdbqt files, opened with the pymol software and visualized. The degree of binding of proteins to molecules was expressed in terms of energy level, and it is generally accepted that the lower the energy, the more stable the conformation of the protein bound to the molecules.

### 4.7. Docking Results Verification Experiments

A double antibody one-step sandwich enzyme-linked immunosorbent assay (ELISA) was used. To the pre-coated wells with enzyme (ALS, DHAD, HPPD) antibodies, specimens, standards and horseradish peroxidase-labeled detection antibodies were added sequentially, incubated and thoroughly washed. The wells were developed with the substrate 3,3’,5,5’-Tetramethylbenzidine (TMB), which was converted to blue by peroxidase and to the final yellow by acid. The absorbance (OD) was measured at 450 nm using an enzyme marker to calculate the sample activity.

### 4.8. Statistical Analysis

Significance was determined by the F-test or Levene’s test. Statistical significance was evaluated using a two-tailed *t*-test (for all two-group comparisons) or one-way analysis of variance (ANOVA) followed by Tukey’s and an LSD test (for multi-group comparisons). Data were presented as mean ± standard error (SE) and the *p*-value < 0.05 was considered statistically significant with * *p* < 0.05 and ** *p* < 0.01. For multi-group comparisons, asterisks were used, respectively. “ns” indicated no significant difference for two-group comparisons. In the case of one-way ANOVA, “NS” was used to indicate no significant difference.

## 5. Conclusions

The eight compounds were first isolated from *B. oleracea* L. Compounds 1–5 are indole skeleton compounds, compound 6 is an aldehyde, compound 7 is a diterpenoid and compound 8 is an organic acid. After evaluating the biological activities of all the compounds against *L. sativa* Linn., *C. album* and *P. miliaceum*, it was found that indole-3-carboxaldehyde, (-)-Spirobrassinin, brassicanate A sulfoxide, methyl 1H-indole-3-carboxylate and Tribenuron Methyl^®^ exhibit comparable inhibitory abilities. The demonstrated biological activities of these compounds offer a compelling explanation for the strong allelopathic effects observed in *B. oleracea* L. Additionally, brassicanate A sulfoxide showed stable binding with targets such as HPPD, DHAD, ALS, PLY2-HAB1 and TIR1. This further confirms that brassicanate A sulfoxide is an ideal candidate molecule in the development of environmentally friendly herbicides.

## Figures and Tables

**Figure 1 plants-12-02576-f001:**
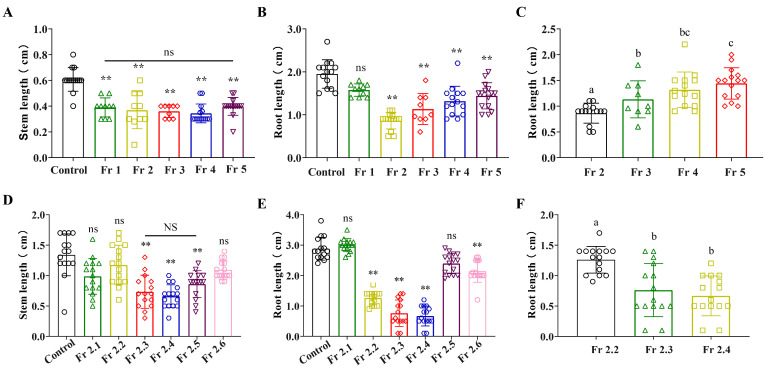
Effect of different fractions of *B. oleracea* L. at 0.25 mg/mL concentration on the growth of *L. sativa* Linn. seedlings in Petri dishes. ** indicates significant differences relative to the control (*p* < 0.01). Lowercase letters indicates significance between fractions (*p* < 0.05). “NS” indicates no significant difference in ANOVA results, “ns” was no significant difference after *t*-test. Each bar graph shows the mean ± standard error. The different symbols in the columns indicate the phenotypic data of *L. sativa* Linn. seedlings after different fraction or subfraction treatments. (**A**) Effect of different fractions on *L. sativa* Linn. stem. (**B**,**C**) Effect of different fractions on *L. sativa* Linn. root. (**D**) Effect of different subfractions on *L. sativa* Linn. stem. (**E**,**F**) Effect of different subfractions on *L. sativa* Linn. root.

**Figure 2 plants-12-02576-f002:**
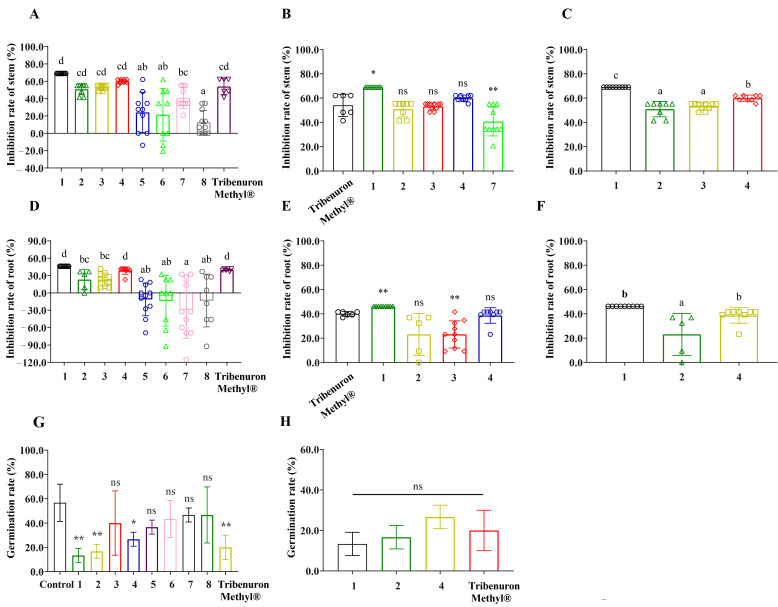
The effects of compounds from *B. oleracea* L. (1–8) and Tribenuron Methyl^®^ on *L. sativa* Linn. germination and seedling growth. The concentrations of the eight compounds found in *B. oleracea* L. and the positive control herbicide, Tribenuron Methyl^®^, were used at a concentration of 0.25 mg/mL. * indicates significant differences relative to the control or positive control (*p* < 0.05), ** indicates significant differences relative to the control or positive control (*p* < 0.01). Lowercase letters indicated significance between fractions (*p* < 0.05). “ns” indicates no significant difference after *t*-test. Each bar graph shows the mean ± standard error. The different symbols in the columns indicate the inhibition rate of *L. sativa* Linn. seedlings after different compound treatments. (**A**–**C**) Inhibition rate of different compounds on *L. sativa* Linn. stem. (**D**–**F**) Inhibition rate of different compounds on *L. sativa* Linn. root. (**G**,**H**) Effect of different compounds on *L. sativa* Linn. germination rate.

**Figure 3 plants-12-02576-f003:**
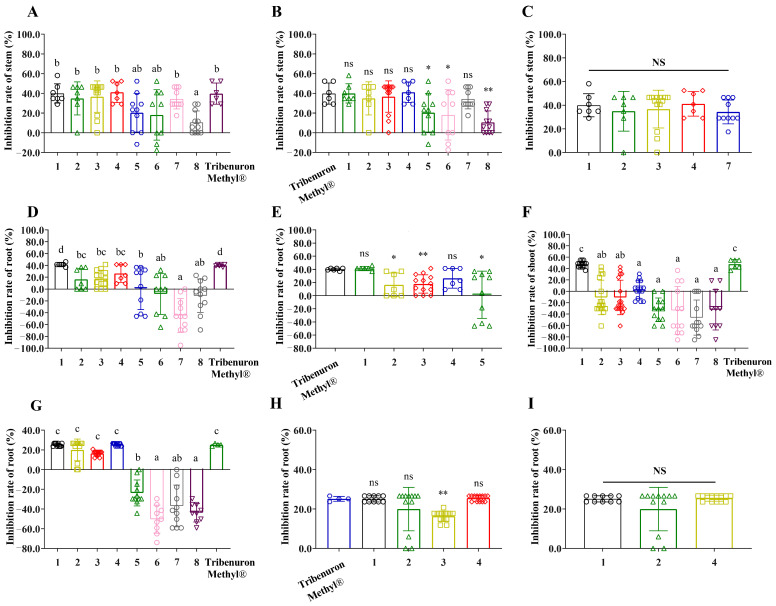
The effects of compounds from *B. oleracea* L. (1–8) and Tribenuron Methyl^®^ on *C. album* and *P. miliaceum* seedling growth. * indicated significant differences relative to the control or positive control (*p* < 0.05), ** indicated significant differences relative to the control or positive control (*p* < 0.01). Lowercase letters indicated significance between fractions (*p* < 0.05). “NS” indicated no significant difference in ANOVA results, “ns” showed no significant difference after *t*-test. Each bar graph was the mean ± standard error. The different symbols in the columns indicate the inhibition rate of *C. album* and *P. miliaceum* seedlings after different compound treatments. (**A**–**C**) Inhibition rate of different compounds on *C. album* stem. (**D**,**E**), Inhibition rate of different compounds on *C. album* root. (**F**) Inhibition rate of different compounds on *P. miliaceum* shoot. (**G**–**I**) Inhibition rate of different compounds on *P. miliaceum* root.

**Figure 4 plants-12-02576-f004:**
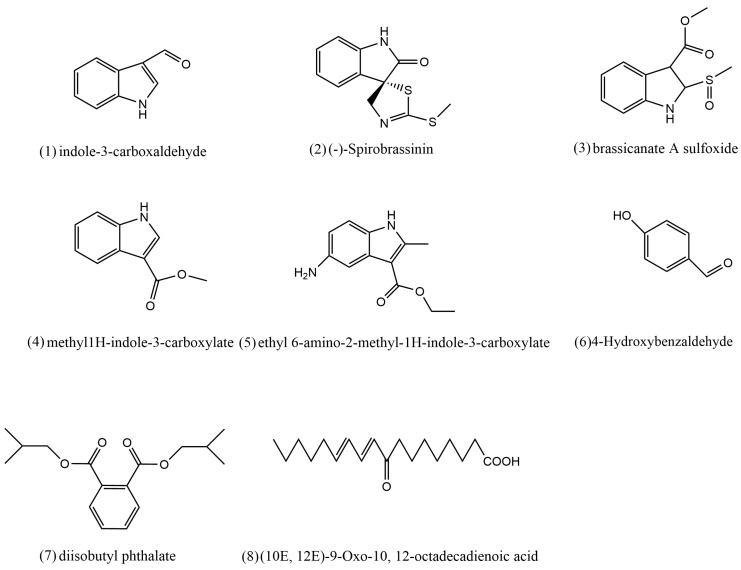
Chemical structures of compounds isolated from *B. oleracea* L. (**1**–**8**).

**Figure 5 plants-12-02576-f005:**
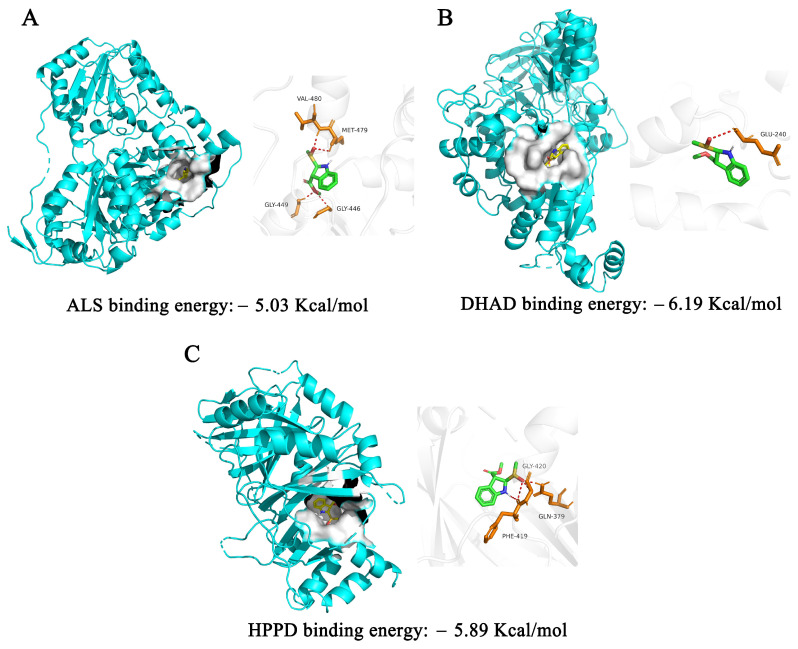
Visualization of molecular docking of brassicanate A sulfoxide with herbicide target proteins. (**A**) Molecular docking of brassicanate A sulfoxide and ALS. (**B**) Molecular docking of brassicanate A sulfoxide and DHAD. (**C**) Molecular docking of brassicanate A sulfoxide and HPPD. Red dashed lines indicate hydrogen bonds.

**Figure 6 plants-12-02576-f006:**
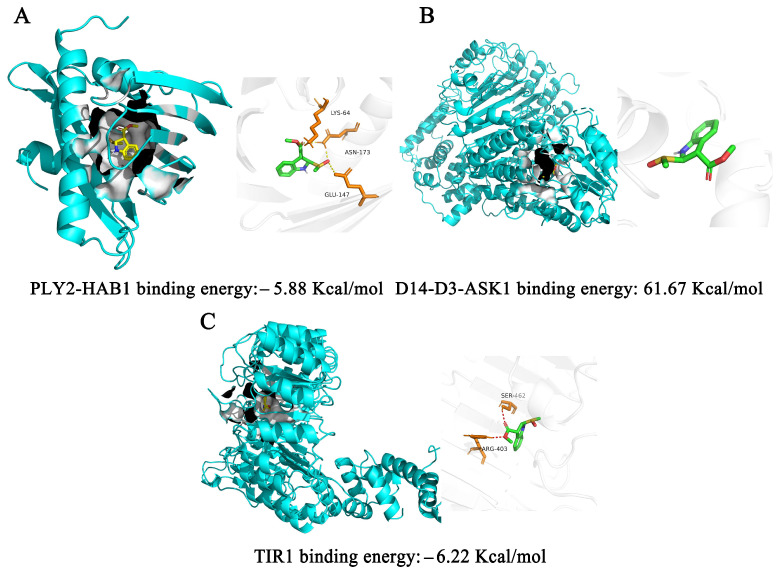
Visualization of molecular docking of brassicanate A sulfoxide with plant growth-regulator. (**A**) Molecular docking of brassicanate A sulfoxide and PLY2-HAB1. (**B**) Molecular docking of brassicanate A sulfoxide and D14-D3-ASK1. (**C**) Molecular docking of brassicanate A sulfoxide and TIR1. Yellow dashed lines indicate hydrogen bonds.

**Figure 7 plants-12-02576-f007:**
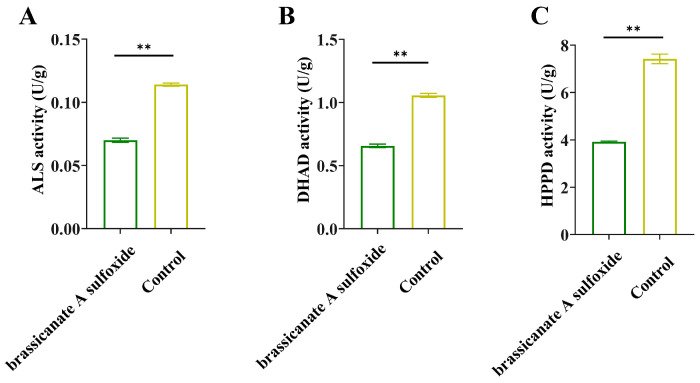
The effects of brassicanate A sulfoxide on ALS, DHAD and HPPD activity. ** indicates significant differences relative to the control (*p* < 0.01). (**A**) Effect of brassicanate A sulfoxide on ALS activity in *L. sativa* Linn. seedlings. (**B**) Effect of brassicanate A sulfoxide on DHAD activity in *L. sativa* Linn. seedlings. (**C**) Effect of brassicanate A sulfoxide on HPPD activity in *L. sativa* Linn. seedlings.

## Data Availability

Data are available within the article or its Appendix A.

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
