# Peer review of "Isolation and Identification of Herbicidal Active Compounds from Brassica oleracea L. and Exploration of the Binding Sites of Brassicanate A Sulfoxide"

_plants, 2023, doi:10.3390/plants12132576_

Round 1

Reviewer 1 Report

The manuscript is of wrathful consideration for weed scientists and agronomists. This is based upon effective control of important weed of that region. The write-up of whole manuscript is up to mark. English quality is acceptable. However, there are few suggestions that need to be incorporated before final acceptance as elaborated below:

-          The topic of the paper should be changed. It’s not giving a clear idea for the reads understanding.

-          The Abstract needs few details about materials of the experiments.

-          Please note the opening paragraph of the introduction could provide stronger context to the paper, and, similarly, the findings at the end could potentially be richer.

-          At the end of Introduction section, there should be clear hypothesis and objectives of the designed study.

-          Statistical designs and data analysis confusing. It needs to present in a proper way

-          Add more information about active ingredients.

-          Cutting conclusion section is an important part in your paper and its better to be added.

Reviewer 2 Report

The manuscript ‘Isolation and Identification of Herbicidal Active Compounds from Brassica oleracea L. var. and Exploration of Their Target Binding Sites' describes the activity-guided isolation of eight compounds from B. oleracea L. with plant growth inhibitory effect. For the most active compound, brassicanate A sulfoxide, a possible mechanism of action was predicted basing upon strong binding to known herbicidal targets and plant growth regulatory targets. The work conforms to the journal scope. The technical quality is good and the bioassays are well conducted. No important references are omitted. Introduction is sufficient to introduce readers to the subject. Results are described in a short and clear way. Experimental part is written well. Isolation and bioassays methodologies are standard and carried out properly. Conclusions are consistent with the facts found and arguments presented.

Nonetheless, there are some issues to be solved by the authors.

General questions.

Is the plant name ‘Brassica oleracea L. var.’ used correctly from the botanical nomenclature point of view? Shouldn’t it be shortened to ‘Brassica oleracea L.’?

2.1. Extraction and separation processes.

The extractant should be noted prior to the fractionation of an extract.

2.2. Isolation of allelochemicals.

The authors had better to indicate briefly how the mentioned 8 compounds were isolated and which subfractions from. The concentrations of the tested compounds and the controls should be provided in the legend to the Fig. 2 as well as used when results are discussed. Otherwise, it is not evident that compound 3 to be the most active one and to be subjected to docking studies. The chemical names should start with lower case letters, i.e. ‘indole-3-carboxaldehyde’ instead of ‘Indole-3-carboxaldehyde’. It is curious if diisobutyl phthalate is a true metabolite or a contamination from the plastic ware used. Have the authors checked the metabolic authencity of it?

2.3. Docking of brassicanate A sulfoxide into the active site of herbicide.

The hydrogen bond is not a type of the covalent bond.

Reviewer 3 Report

The introduction should provide more data from the specialized literature regarding the allelopathic potential of the species

The synthetic presentation of the results should be included in the abstract, not in the introduction

The determination by chromatography is briefly presented, also in the material and method, the species on which the testing is done are not presented

The presented results are supported by a low number of bibliographic references

Round 2

Reviewer 1 Report

Authors revised the paper according to my suggestions.